# Continued high rates of antibiotic prescribing to adults with respiratory tract infection: survey of 568 UK general practices

Martin C Gulliford,[1] Alex Dregan,[1] Michael V Moore,[2] Mark Ashworth,[1] Tjeerd van Staa,[3,4] Gerard McCann,[3] Judith Charlton,[1] Lucy Yardley,[2] Paul Little,[2] Lisa McDermott[1]

[1]King's College London, Primary Care and Public Health Sciences, London, UK
[2]Department of Primary Care and Population Sciences, University of Southampton, Southampton, UK
[3]Clinical Practice Research Datalink (CPRD) Division, Medicines and Healthcare Products Regulatory Agency, London, UK
[4]Health eResearch Centre, Farr Institute for Health Informatics Research, University of Manchester, London, UK

**Correspondence to**
Dr Lisa McDermott;
lisa.mcdermott@kcl.ac.uk

## ABSTRACT

**Objectives:** Overutilisation of antibiotics may contribute to the emergence of antimicrobial drug resistance, a growing international concern. This study aimed to analyse the performance of UK general practices with respect to antibiotic prescribing for respiratory tract infections (RTIs) among young and middle-aged adults.

**Setting:** Data are reported for 568 UK general practices contributing to the Clinical Practice Research Datalink.

**Participants:** Participants were adults aged 18–59 years. Consultations were identified for acute upper RTIs including colds, cough, otitis-media, rhino-sinusitis and sore throat.

**Primary and secondary outcome measures:** For each consultation, we identified whether an antibiotic was prescribed. The proportion of RTI consultations with antibiotics prescribed was estimated.

**Results:** There were 568 general practices analysed. The median general practice prescribed antibiotics at 54% of RTI consultations. At the highest prescribing 10% of practices, antibiotics were prescribed at 69% of RTI consultations. At the lowest prescribing 10% of practices, antibiotics were prescribed at 39% RTI consultations. The median practice prescribed antibiotics at 38% of consultations for 'colds and upper RTIs', 48% for 'cough and bronchitis', 60% for 'sore throat', 60% for 'otitis-media' and 91% for 'rhino-sinusitis'. The highest prescribing 10% of practices issued antibiotic prescriptions at 72% of consultations for 'colds', 67% for 'cough', 78% for 'sore throat', 90% for 'otitis-media' and 100% for 'rhino-sinusitis'.

**Conclusions:** Most UK general practices prescribe antibiotics to young and middle-aged adults with respiratory infections at rates that are considerably in excess of what is clinically justified. This will fuel antibiotic resistance.

## INTRODUCTION

Overuse of antibiotic drugs is leading to increasing antimicrobial drug resistance. As there are now fewer new antibiotic drugs

### Strengths and limitations of this study

- The findings are derived from a large, representative sample of UK general practices.
- Findings did not include information concerning severity of illness or the presence of comorbidity, which might have accounted for the prescription of antibiotics in some cases.
- Only prescriptions issued by the practice were analysed, as it was not possible to estimate from electronic health records whether prescriptions were dispensed, or whether a delayed prescribing strategy was intended.

being developed, it is important to preserve the effectiveness of presently available antibiotics for future generations.[1] The Chief Medical Officer's annual report for 2011[1] promoted the concept of antimicrobial stewardship, which means that unnecessary or inappropriate use of antibiotics should be avoided so as to minimise the selection of antibiotic resistant strains of organisms. In addition to increasing antimicrobial drug resistance, the overuse of antibiotic drugs can lead to unnecessary side effects and increase future consultations for respiratory tract infections (RTIs).[2 3] In primary care, RTIs are a common reason for consultation and antibiotics are frequently prescribed. RTIs account for about 60% of antibiotic prescribing in primary care.[4] Previous studies showed that antibiotic utilisation at consultations for respiratory infections declined during the 1990s but has remained constant since.[5] However, there has been a long-term decline in the rate of consultation for RTI in UK primary care.[6]

In 2008, the National Institute for Health and Care Excellence (NICE) recommended that most acute RTIs, including colds, coughs,

sore throats, otitis-media and rhino-sinusitis, could be managed without antibiotics and recommended that either a 'no antibiotic' or 'delayed antibiotic prescribing' strategy should be agreed for most patients.[4]

We recently completed a large cluster randomised trial to reduce antibiotic prescribing among general practices that contribute to the Clinical Practice Research Datalink (CPRD).[7][8] The study aimed to reduce unnecessary antibiotic prescribing using an electronically delivered intervention. The present analysis included data from general practices that participated in the trial as well as data from non-trial general practices. We aimed to describe the performance of UK general practices with respect to antibiotic prescribing for respiratory illness in young and middle-aged adults.

## METHODS
The UK CPRD provided the data source for the study. The CPRD is a database of prospectively collected electronic medical records from approximately 7% of UK general practices. It includes records for all prescriptions issued and medical diagnoses recorded.[9] The study included all CPRD general practices that were included in the cluster trial,[8] as well as sample data for all CPRD general practices that were not included in the trial. All registered patients were included for the trial practices and, in order to provide a manageable data set for analysis, a random sample of registered patients was taken from non-trial practices. The period of study included the 12 months preceding the start date of the cluster trial with the date of random allocation was used as the index date.[8] The practices were allocated in five batches between 26 November 2010 and 26 April 2011. For non-trial practices, the median of the allocation dates, 20 January 2011, was used as the index date.

Individual participants were adults aged 18–59 years. This was consistent with the eligibility criteria for the trial,[7] which aimed to exclude children and older adults who might be at higher risk of complications. For each participant, we analysed their clinical record for 12 months before the trial index date. General practices were analysed as a single group as there were no overall baseline differences between trial and non-trial practices with respect to consultation and antibiotic prescribing rates.[8] The analysis used 232 general practice Read medical codes (recorded by general practitioners for each patient who consulted with a RTI), including those for 'colds' and 'upper respiratory tract infection' (URTI); 'cough' and 'bronchitis'; 'sore throat', including pharyngitis, laryngitis, tracheitis, epiglottitis and tonsillitis; 'otitis-media' including acute otitis-media and otitis-media; and 'rhino-sinusitis' including all forms of sinusitis. These were used to identify consultations for acute RTIs. The source of information for RTI consultations was represented by clinical, referral and test files data. Only first consultations within an episode were included using a 10-day time window. Therapy file data were used to ascertain antibiotic prescribing information. Antibiotic prescriptions were identified using drug codes that map to section 5.1 of the British National Formulary, excluding drugs used to tuberculosis and leprosy. For each general practice, we estimated the consultation rate for RTI per 1000 registered patients, the antibiotic prescribing rate per 1000 registered patients and the proportion of RTI consultations with antibiotics prescribed as reported previously.[6]

## RESULTS
The selection of general practices and patients into the analysis is outlined in figure 1. There were 582 CPRD

**Figure 1** Flow chart showing selection of general practices and participants ( RTI, respiratory tract infection; CPRD, Clinical Practice Research Datalink).

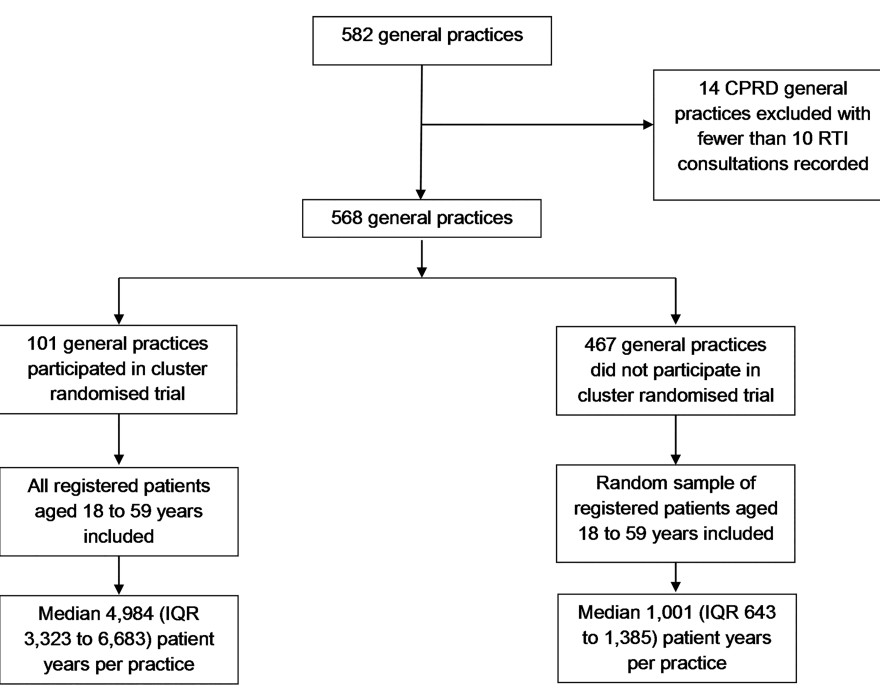

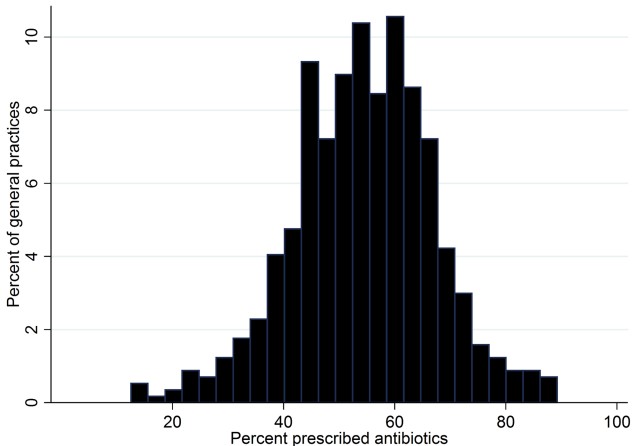

**Figure 2** Distribution for per cent of respiratory tract infection consultations with antibiotics prescribed for adults aged 18–59 years at 568 UK general practices.

general practices available for analysis, 14 practices which contributed fewer than 10 RTI consultations during the study period were excluded leaving 568 for further analysis, including 101 that participated in the trial and 467 that did not participate in the trial. There were 431 practices in England, 21 in Northern Ireland, 66 in Scotland and 50 in Wales. Data were analysed for registered patients aged 18–59 years. There were 1 016 779 registered patients with 219 162 consultations for RTI and 118 583 antibiotic prescriptions available for analysis. There was a mean rate of 217 RTI consultations per 1000 person years and a mean rate of 119 antibiotic prescriptions for RTI per 1000 person years. Coefficients of variation of the practice-specific rates were 0.30 for the RTI consultation rate and 0.41 for the antibiotic prescribing rate, respectively.

Figure 2 shows the distribution of the practice-specific proportion of RTI consultations with antibiotics prescribed for 568 UK general practices. Considering all RTI consultations as a single group, most practices prescribed antibiotics at between 30% and 80% of RTI consultations. There were only 18 (3%) of practices that prescribed antibiotics at fewer than 30% of RTI

consultations and 4 (1%) of practices that prescribed antibiotics at fewer than 20% of RTI consultations.

Table 1 shows the distribution of the practice-specific prescribing proportions according to the type of RTI consultation. The figures represent the per cent of RTI consultations with antibiotics prescribed for the general practice that occupies the stated position in the distribution of results for all 568 practices. The median practice prescribed antibiotics at 54% of RTI consultations. The highest prescribing 10% of practices issued prescriptions at 69% or more of RTI consultations, and the highest prescribing 5% of practices issued prescriptions at 74% or more of all RTI consultations. By contrast, the lowest prescribing 10% of practices issued prescriptions at 39% of RTI consultations, and the lowest prescribing 5% of practices issued antibiotic prescriptions at 33% of RTI consultations.

Consultations for 'cough and bronchitis' accounted for 39% of RTI consultations; 'sore throat' 27%; 'colds and URTI' 19%; 'rhino-sinusitis' 9%; and 'otitis media' 6%. Table 1 shows the distribution of practice-specific prescribing proportions according to the type of RTI consultation. The median practice issued antibiotic prescriptions at 38% of consultations for 'colds', 48% for 'cough', 60% for 'otitis media' and 'sore throat', and 91% for 'rhino-sinusitis'. However, the highest prescribing 10% of practices issued antibiotic prescriptions at 72% of consultations for 'colds and URTI', 67% for 'cough and bronchitis', 78% for 'sore throat', 90% for 'otitis-media' and 100% for 'rhino-sinusitis'. The lowest prescribing 10% of practices issued antibiotic prescriptions at 14% of consultations for 'colds and URTI', 28% for 'cough' and 41% for 'sore throat'.

## DISCUSSION

National guidance in the UK recommends that most patients presenting with acute RTIs can be managed with either no antibiotic prescribing or delayed antibiotic prescribing, with a prescription only being used if symptoms do not improve.[4] The present results show that most general practices in the UK depart substantially from recommended standards of good practice

**Table 1** Centiles of the distribution of the proportion (%) of RTI consultations with antibiotics prescribed at 568 UK general practices

| | Proportion of RTI consultations with antibiotics prescribed at UK general practices | | | | | | |
|---|---|---|---|---|---|---|---|
| | Lowest 5% of practices | Lowest 10% | Lowest 25% | Median | Highest 25% | Highest 10% | Highest 5% of practices |
| All | 33 | 39 | 46 | 54 | 63 | 69 | 74 |
| Colds and URTI | 9 | 14 | 25 | 38 | 56 | 72 | 81 |
| Cough and bronchitis | 22 | 28 | 38 | 48 | 59 | 67 | 71 |
| Otitis-media | 22 | 32 | 45 | 60 | 75 | 90 | 100 |
| Rhino-sinusitis | 67 | 75 | 83 | 91 | 98 | 100 | 100 |
| Sore throat | 35 | 41 | 50 | 60 | 68 | 78 | 83 |

Note that one practice may not occupy the same centile of prescribing for each condition.
URTI, upper respiratory tract infection.

with respect to antibiotic prescribing in a generally low-risk age range of young and middle-aged adults. Even for common colds and URTIs, which are generally acknowledged to have a viral aetiology, antibiotics may be prescribed for a third of patients overall and for more than 80% of patients at some general practices. A number of trials have now shown that antibiotic prescribing may be reduced through educational interventions, together with feedback of prescribing information.[10–12] However, these interventions generally have modest effects with generally less than 10–15% reduction in antibiotic prescribing. As Linder[13] has observed, current antibiotic prescribing appears to be 'way off the mark' when viewed in the context of systematic review evidence of lack of benefit[14] and current recommendations for good clinical practice.[4]

Our study had the strengths of a large, representative sample of UK general practices. We acknowledge that we did not include information concerning severity of illness or the presence of comorbidity, which might have accounted for the prescription of antibiotics in some cases. We only analysed prescriptions issued by the practice and it was not possible to estimate from electronic health records whether the prescription was dispensed, or whether a delayed prescribing strategy was intended. There is a Read code for deferred antibiotic therapy (8BP0.00) but this was recorded for fewer than 0.5% of medical events. It is unlikely that delayed prescribing can fully account for the high prescribing rates. In an observational study in 13 000 adults with sore throat, immediate antibiotics were issued in 42% and 12% given delayed antibiotics.[15] Delayed prescribing is unlikely to vitiate our conclusion that most UK practices prescribe antibiotics to excess. One driver of prescribing is worries about complications but complications are hard to predict and rare and that delayed prescribing is probably as effective as immediate prescribing to reduce the risk of complications.[12] The categories used for analysis may have combined several different entities, for example, prescribing may be more frequent for cases coded as 'bronchitis' than for 'cough'. Prescribing for sinusitis was generally high, even at lower prescribing practices. We have not analysed practice characteristics as possible predictors of antibiotic prescribing, but such analyses typically only explain a small proportion of the variation between practices.[16] The results suggest that most practices commonly prescribe antibiotics unnecessarily. Patient characteristics such as age,[17] gender, comorbidity, smoking status or deprivation category might also be associated with prescribing decisions. Nevertheless, these results suggest that many patients may be prescribed antibiotics unnecessarily. Reducing antibiotic prescribing may lead to lower consultation rates for RTI.[18] The present study did not include children who represent some of the highest users of antibiotic prescriptions[17] but children will be included in a planned cluster randomised trial in CPRD to start in 2015.

The present results have implications for communications with the public as well as for practice prescribing policies. Respiratory infections in this age group are both self-limiting and carry a low risk of complications, moreover the impact of antibiotics on symptom severity and duration is at best marginal. Respiratory infections may be better managed through patient self-care rather than primary care consultation. The high rates of antibiotic prescribing reported by this study indicate a need to shift the entire distribution for antibiotic prescribing to lower levels, since there are very few practices that are not prescribing antibiotics to excess, fuelling the development of antibiotic resistance. In addition, there are immediate direct costs from prescribing antibiotics, as well as risks of drug side effects and the perpetuation of unnecessary consultation patterns. There needs to be an active professional debate concerning an overall level of antibiotic utilisation for RTI that might be acceptable, and the size of reduction that individual practices should aim to achieve as a matter of urgency.

**Contributors** MCG designed, supervised and drafted the paper. LM assisted with draft and conclusions. JC and AD contributed to data analysis. MA, MVM, PL, LY contributed to design, write-up and interpretation of data. TvS and GM contributed to practice recruitment and facilitated access to CPRD. All authors contributed to the paper and approved the final draft. All authors read and approved the final manuscript.

**Funding** The study was supported by the Joint Initiative in Electronic Patient Records and Databases in Research, a partnership between the Wellcome Trust, Medical Research Council, Economics & Social Research Council and Engineering & Physical Sciences Research Council (WT086129MA). MCG and AD were supported by the National Institute for Health Research (NIHR) Biomedical Research Centre at Guy's and St Thomas' NHS Foundation Trust and King's College London. CPRD has received funding from the MHRA, Wellcome Trust, Medical Research Council, NIHR Health Technology Assessment programme, Innovative Medicine Initiative, UK Department of Health, Technology Strategy Board, Seventh Framework Programme EU, various universities, contract research organisations and pharmaceutical companies.

**Competing interests** None.

**Ethics approval** The study obtained ethical approval from the South West London Research and Ethics Committee (09/H0806/81) and was approved by the CPRD Independent Scientific Advisory Committee (ISAC protocol 08_083).

**Provenance and peer review** Not commissioned; externally peer reviewed.

**Data sharing statement** No additional data are available.

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
