## [Reviewer comments · BMJ Open]

Some articles will have been accepted based in part or entirely on reviews undertaken for other BMJ Group journals. These will be reproduced where possible.

ARTICLE DETAILS

TITLE (PROVISIONAL)	Continued high rates of antibiotic prescribing to adults with respiratory tract infection. Survey of 568 UK general practices
AUTHORS	McDermott, Lisa; Gulliford, Martin; Dregan, Alexandru; Moore, Michael; Asthworth, Mark; van Staa, Tjeerd; McCann, Gerard; Charlton, Judith; Yardley, Lucy; Little, Paul

VERSION 1 - REVIEW

REVIEWER	Katrin Gaardbo Kuhn Epidemiologist (PhD) Statens Serum Institut Copenhagen, Denmark
REVIEW RETURNED	22-Aug-2014

GENERAL COMMENTS	The authors have submitted a very well-written paper which addresses an issue that is currently of interest to both the general public as well as specialised scientific audience. The target of respiratory infections is well-chosen and the sample size is adequate. My concerns with the paper are mainly related to the patient/practice recruitment and the presentation of the results. Please see below. Methods 1. Randomisation procedure could do with some elaboration.2. The number of practices involved/participating and how they were identified is confusing to determine (see Results) Results This section in my opinion needs significant revision. 1. I had difficulty determining how many practices/patients were selected and from where; 104 from the original trial and the rest 'random samples of patients from practices that did not participate'? If the 'rest' are individual patients rather than 'whole' practices (all 18-59 y.o) please consider this in relation to your data. Suggest to show a flow diagram of recruitment.2. Table 2 was not supplied with my version of the paper. It is referred to on p7, l 21. From the numbers listed in lines 22 onwards, I assume you are referring to Table 1 and 'Table 2' is just a typo. This section is overall hard to read. I would
---

	suggest to re-write it.  3. Need to present data to show why the highest/lowest GPs are prescribing the way they are. Is it related to the patient population or area (see below)? If there are significant differences in the patient population between the GPs, you will need to take this into consideration. 4. The results could benefit from also focusing on patient data rather than just practice data (see below). This would also make better sense if data collected do not represent the whole practice (18-59 y.o) but rather individual patients (see point 1 above). Discussion/General comments The Discussion leaves me with many questions that I think can be answered by the results the authors have collected: Is there a demographic reason why the high/low prescribing practices are like that? What are the ‘problem’ patient groups (age, gender) where we can target future campaigns? The conclusion about ‘an active professional debate’ seems rather vague without these specific details. Therefore:  - The difference between low/high prescribing practices is striking. I doubt that this is only related to the individual doctors. I miss some basic data on how well-matched the GPs were with respect to patient population covered: average age (in particular), area covered (in terms of socio-economic factors) etc. - The age interval included is pretty large. Would be nice to see patient-specific data at a higher resolution; are some age groups more likely to be prescribed antibiotics for RI, what about gender... etc. Combined with the info about patient population, this may highlight why the ‘highest practices’ are high. Given that by far the most antibiotics for RI are prescribed to children, I am surprised that no data have been collected for this group. I fear that if we repeat this study in Denmark, we would see a huge over-prescription in children, particularly for RIs – mostly related to parental pressure. I realise that these data are not feasible to present now, given the original study design, but it would be nice to mention in the Discussion.
--	---

REVIEWER	Michael Grover, DO Assistant Professor Mayo Clinic College of Medicine Chair, Department of Family Medicine Mayo Clinic, Scottsdale Arizona, USA
REVIEW RETURNED	27-Aug-2014

GENERAL COMMENTS	The authors describe their examination of antibiotic utilization at 568 UK General Practices contributing to the Clinical Practice Research Network. The sample was limited to patients who were 18-59 years of age in order to decrease the probability of including those with co-morbidities and those at increased risk of complications. They found a median antibiotic use rate of 54% for acute respiratory tract infection patients. Rates range from a low of 38% for colds/URI patients to 91% for acute rhinosinusitis patients. This analysis of clinical record data provides further evidence to the continued pattern of antibiotic overuse. This overuse is justified by the authors as a problems due to the potential development of bacterial resistance, which is certainly evident. However, there are a myriad of other reasons to justify the evaluation of this issue including the associated costs from unnecessary treatment, the potential development of side effect or complication related to treatment and providing fuel to the false belief of a bacterial etiology for the ARTI illness (which is usually viral) and support to the continued use of them in the future. The authors might consider expanding the discussion of other reasons to address antibiotic overuse beyond bacterial resistance. A minor point- the combining of "cough and bronchitis" diagnoses may have artificially lowered the antibiotic use rates as most reports of treatment for acute bronchitis (acute cough illness/"chest colds") are substantially higher. It is also notable that in the practices with the lowest overall prescribing rates the use of antibiotics for sinusitis is still substantial, 67%. This is an area of concern for our collective attention.
---

REVIEWER	Yali Zhao, Associate Professor; Aimin Guo, Professor Capital Medical University, Beijing, China. No competing interests exist.
REVIEW RETURNED	28-Aug-2014

GENERAL COMMENTS	10. more data should be presented in the results for addressing fully research objective. Regardless of the following comments, this is a timely study which provides useful and practical evidence to the performance of UK general practices in antibiotic prescribing for RTI in young and middle-aged adults. 1. In the Introduction section, In introduction, why the cluster randomized trial was mentioned? (line 39) Please kindly provide the reasons. Please briefly introduce the protocol of cluster randomized trials, including objective, design, practices, intervention, outcome, and analysis using one or two sentences in method part. Line 43: "we aimed to describe the performance..." In abstract, line 5-6, the authors mentioned "This study aimed to evaluate the
---

	performance..."There seems no evaluation, only description. 2. In the methodology section, (1) The authors did not mention the number of non-trial practices included in the study. (2) Based on the result section, besides person years, the number of practices and consultations should be provided and described in detail. (3) Please interpret the meaning of the "232 Read codes". (4) A brief flow chart maybe a better way to illustrate how the study participants, e.g., trial practices, non-trial practices, consultations, and person years, were drawn. It will support the opinion that "Our study had the strengths of a large, representative sample of UK general practices" in the discussion section. 3. In the result section, (1) "There were 582 practices available for analysis." – on page 6. It is not quite clear how the authors get the data. In methodology section, only 104 practices in the cluster trial were described. (2) Refer to the description on page 6 para 2, it would be better to show a comparative table that describe both similarities and differences between trial practices and non-trial practices, e.g., consultation rate, antibiotic prescribing rate, and the proportion of RTI consultations with antibiotics prescribed, to help readers have a clear idea. (3) Table 2 was mentioned on page 7 para 2, but I did not find the table at the end of the manuscript. 4. In the discussion section, (1) It was mentioned that "Even for common colds and upper respiratory tract infections, which are generally acknowledged to have a viral aetiology, antibiotics may be prescribed for a third of patients overall and for more than 80% of patients at some general practices" (on page 7-8). However, the result section did not show the data to support the above viewpoint.
--	--

VERSION 1 – AUTHOR RESPONSE

Reviewer: 1

The authors have submitted a very well-written paper which addresses an issue that is currently of interest to both the general public as well as specialised scientific audience. The target of respiratory infections is well-chosen and the sample size is adequate. My concerns with the paper are mainly related to the patient/practice recruitment and the presentation of the results. Please see below.

Thank you for this feedback.

1. Randomisation procedure could do with some elaboration.

Thank you, we have added a reference to the trial report (reference 6). However, as the period of analysis is exclusively before randomisation, we have not added further details here, as this information may be confusing.

2. The number of practices involved/participating and how they were identified is confusing to determine (see Results)

Thank you, we now explain (page 5): 'The UK Clinical Practice Research Datalink (CPRD) provided the data source for the study. The CPRD is a database of prospectively collected electronic medical records from approximately 7% of UK general practices. It includes records for all prescriptions issued and medical diagnoses recorded.(7) ...The study included all CPRD general practices that were included in the cluster trial, as well as sample data for all CPRD general practices that were not included in the trial. All registered patients were included for the trial practices that participated in the trial and, in order to provide a manageable dataset for analysis, a random sample of registered patients was taken from non-trial practices that did not participate in the trial.'

1. I had difficulty determining how many practices/patients were selected and from where; 104 from the original trial and the rest 'random samples of patients from practices that did not participate'? If the 'rest' are individual patients rather than 'whole' practices (all 18-59 y.o) please consider this in relation to your data. Suggest to show a flow diagram of recruitment.

Thank you, we have now added a flowchart as the new Figure 1. We explain this in the text (page 6): 'The selection of general practices and patients into the analysis is outlined in Figure 1. There were 582 CPRD general practices available for analysis, 14 practices which contributed fewer than 10 RTI consultations during the study period were excluded leaving 568 for further analysis, including 101 that participated in the trial and 467 that did not participate in the trial. There were 431 practices in England, 21 in Northern Ireland, 66 in Scotland and 50 in Wales.'

2. Table 2 was not supplied with my version of the paper. It is referred to on p7, l 21. From the numbers listed in lines 22 onwards, I assume you are referring to Table 1 and 'Table 2' is just a typo. This section is overall hard to read. I would suggest to re-write it.

Thank you, this was an error, we now say (page 7): 'Table 1 shows the distribution of practice specific prescribing proportions according to the type of RTI consultation.'

3. Need to present data to show why the highest/lowest GPs are prescribing the way they are. Is it related to the patient population or area (see below)? If there are significant differences in the patient population between the GPs, you will need to take this into consideration.

Thank you, we now comment (page 9): 'We have not analysed practice characteristics as possible predictors of antibiotic prescribing, but such analyses typically only explain a small proportion of the variation between practices. The results suggest that most practices commonly prescribe antibiotics unnecessarily.'

4. The results could benefit from also focusing on patient data rather than just practice data (see below). This would also make better sense if data collected do not represent the whole practice (18-59 y.o) but rather individual patients (see point 1 above).

Thank you, we now comment (page 9): 'Patient characteristics such as age, gender, comorbidity,

smoking status or deprivation category might also be associated with prescribing decisions. Nevertheless, these results suggest that many patients may be prescribed antibiotics unnecessarily.'

The Discussion leaves me with many questions that I think can be answered by the results the authors have collected: Is there a demographic reason why the high/low prescribing practices are like that? What are the 'problem' patient groups (age, gender) where we can target future campaigns? The conclusion about 'an active professional debate' seems rather vague without these specific details.

Thank you for these comments. Our message is that 'The high rates of antibiotic prescribing reported by this study indicate a need to shift the entire distribution for antibiotic prescribing to lower levels, since there are very few practices that are not prescribing antibiotics to excess, fuelling the development of antibiotic resistance.' (page 9). This paper provides evidence on the size of the problem, which we hope professional bodies will debate and take active steps to remedy.

- The difference between low/high prescribing practices is striking. I doubt that this is only related to the individual doctors. I miss some basic data on how well-matched the GPs were with respect to patient population covered: average age (in particular), area covered (in terms of socio-economic factors) etc.
- The age interval included is pretty large. Would be nice to see patient-specific data at a higher resolution; are some age groups more likely to be prescribed antibiotics for RI, what about gender... etc. Combined with the info about patient population, this may highlight why the 'highest practices' are high.

Please see responses to points 3 and 4 above. We think that analyses aiming to explain variations may dilute our message by suggesting that there may be a 'rational' explanation.

Given that by far the most antibiotics for RI are prescribed to children, I am surprised that no data have been collected for this group. I fear that if we repeat this study in Denmark, we would see a huge over-prescription in children, particularly for RIs – mostly related to parental pressure. I realise that these data are not feasible to present now, given the original study design, but it would be nice to mention in the Discussion.

Thank you, we now add (page 9): 'The present study did not include children who represent some of the highest users of antibiotic prescriptions¹⁵ but children will be included in a planned cluster randomised trial in CPRD to start in 2015.'

Reviewer: 2

There are a myriad of other reasons to justify the evaluation of this issue including the associated costs from unnecessary treatment, the potential development of side effect or complication related to treatment and providing fuel to the false belief of a bacterial etiology for the ARTI illness (which is usually viral) and support to the continued use of them in the future. The authors might consider expanding the discussion of other reasons to address antibiotic overuse beyond bacterial resistance.

Thank you we now add (page 4): 'In addition to increasing antimicrobial drug resistance, the over use of antibiotic drugs can lead to unnecessary side-effects and increase future consultations for RTI's.(2,3)'

We also add (page 9): 'In addition, there are immediate direct costs from prescribing antibiotics, as

well as risks of drug side effects and the perpetuation of unnecessary consultation patterns.'

A minor point- the combining of 'cough and bronchitis' diagnoses may have artificially lowered the antibiotic use rates as most reports of treatment for acute bronchitis (acute cough illness/'chest colds') are substantially higher.

Thank you, we now add (page 9): 'The categories used for analysis may have combined several different entities, for example, prescribing may be more frequent for cases coded as 'bronchitis' than for 'cough'.'

It is also notable that in the practices with the lowest overall prescribing rates the use of antibiotics for sinusitis is still substantial, 67%. This is an area of concern for our collective attention.

Thank you, we now add (page 9): 'Prescribing for sinusitis was generally high, even at lower prescribing practices.'

Reviewer: 3

Regardless of the following comments, this is a timely study which provides useful and practical evidence to the performance of UK general practices in antibiotic prescribing for RTI in young and middle-aged adults.

Thank you for this feedback.

1. In the Introduction section, In introduction, why the cluster randomized trial was mentioned? (line 39) Please kindly provide the reasons. Please briefly introduce the protocol of cluster randomized trials, including objective, design, practices, intervention, outcome, and analysis using one or two sentences in method part.

Thank you, we now add (page 4): 'We recently completed a large cluster randomised trial to reduce antibiotic prescribing among general practices that contribute to the Clinical Practice Research Datalink (CPRD).^{5, 6}The study aimed to reduce unnecessary antibiotic prescribing using an electronically delivered intervention. Data from practices included in this trial were also included in the present analysis.' We have references the study protocol and main results (references 7 and 8).

Line 43: 'we aimed to describe the performance...' In abstract, line 5-6, the authors mentioned 'This study aimed to evaluate the performance...' There seems no evaluation, only description.

Thank you, this has been amended: 'This study aimed to analyse the performance of UK general practices with respect to antibiotic prescribing for respiratory tract infections among young and middle-aged adults.'

2. In the methodology section,

(1) The authors did not mention the number of non-trial practices included in the study.

Thank you, this has now been clarified in the flowchart (Figure 1) and associated text.

(2) Based on the result section, besides person years, the number of practices and consultations should be provided and described in detail.

Thank you, we now add (page 6): 'There were 1,016,779 registered patients with 219,162 consultations for RTI and 118,583 antibiotic prescriptions available for analysis.'

(3) Please interpret the meaning of the '232 Read codes'.

Thank you, we now add (pages 5-6): 'The analysis used 232 general practice Read medical codes (recorded by GPs for each patient who consulted with a RTI), including those for 'colds' and 'upper respiratory tract infection' (URTI); 'cough' and 'bronchitis'; 'sore throat', including pharyngitis, laryngitis, tracheitis, epiglottitis and tonsillitis; 'otitis-media' including acute otitis media and otitis media; and 'rhino-sinusitis' including all forms of sinusitis.'

(4) A brief flow chart maybe a better way to illustrate how the study participants, e.g., trial practices, non-trial practices, consultations, and person years, were drawn. It will support the opinion that 'Our study had the strengths of a large, representative sample of UK general practices' in the discussion section.

Thank you, we have now added a flowchart as Figure 1.

3. In the result section,

(1) 'There were 582 practices available for analysis.' – on page 6. It is not quite clear how the authors get the data. In methodology section, only 104 practices in the cluster trial were described.

Thank you, we have now added this detail in the methods and results sections as outlined in our response to Reviewer 1.

(2) Refer to the description on page 6 para 2, it would be better to show a comparative table that describe both similarities and differences between trial practices and non-trial practices, e.g., consultation rate, antibiotic prescribing rate, and the proportion of RTI consultations with antibiotics prescribed, to help readers have a clear idea.

Thank you we now refer to our trial report and say (page 5): 'General practices were analysed as a single group as there were no overall baseline differences between trial and non-trial practices with respect to consultation and antibiotic prescribing rates. (8)'

(3) Table 2 was mentioned on page 7 para 2, but I did not find the table at the end of the manuscript.

Thank you, this has now been amended to read table 1.

4. In the discussion section,

(1) It was mentioned that 'Even for common colds and upper respiratory tract infections, which are generally acknowledged to have a viral aetiology, antibiotics may be prescribed for a third of patients overall and for more than 80% of patients at some general practices' (on page 7-8). However, the result section did not show the data to support the above viewpoint.

Thank you, this refers to the second row of Table 1, which reads:
'Colds' and 'URTI' 9 14 25 38 56 72 81

VERSION 2 – REVIEW

REVIEWER	Michael Grover, DO
----------	--------------------

	Department of Family Medicine Mayo Clinic Scottsdale Arizona, USA
REVIEW RETURNED	27-Sep-2014

GENERAL COMMENTS	the authors were responsive to the critiques of the reviewers and the resulting manuscript has benefited from this. Congratulations on a substantial contribution to the literature in this area.
---

REVIEWER	Katrin Gaardbo Kuhn Statens Serum Institut Denmark
REVIEW RETURNED	03-Oct-2014

GENERAL COMMENTS	Following the revision by the authors, I have no further suggestions for revision. The manuscript is well written and constitutes an interesting contribution to the current debate on antibiotic consumption levels and its implication for public health.
---